# Parametric Task Learning

**Ichiro Takeuchi**
Nagoya Institute of Technology
Nagoya, 466-8555, Japan
takeuchi.ichiro@nitech.ac.jp

**Tatsuya Hongo**
Nagoya Institute of Technology
Nagoya, 466-8555, Japan
hongo.mllab.nit@gmail.com

**Masashi Sugiyama**
Tokyo Institute of Technology
Tokyo, 152-8552, Japan
sugi@cs.titech.ac.jp

**Shinichi Nakajima**
Nikon Corporation
Tokyo, 140-8601, Japan
nakajima.s@nikon.co.jp

## Abstract

We introduce an extended formulation of multi-task learning (MTL) called *parametric task learning (PTL)* that can systematically handle infinitely many tasks parameterized by a continuous parameter. Our key finding is that, for a certain class of PTL problems, the path of the optimal task-wise solutions can be represented as piecewise-linear functions of the continuous task parameter. Based on this fact, we employ a parametric programming technique to obtain the common shared representation across all the continuously parameterized tasks. We show that our PTL formulation is useful in various scenarios such as learning under non-stationarity, cost-sensitive learning, and quantile regression. We demonstrate the advantage of our approach in these scenarios.

## 1   Introduction

Multi-task learning (MTL) has been studied for learning multiple related tasks simultaneously. A key assumption behind MTL is that there exists a common shared representation across the tasks. Many MTL algorithms attempt to find such a common representation and at the same time to learn multiple tasks under that shared representation. For example, we can enforce all the tasks to share a common feature subspace or a common set of variables by using an algorithm introduced in [1, 2] that alternately optimizes the shared representation and the task-wise solutions.

Although the standard MTL formulation can handle only a finite number of tasks, it is sometimes more natural to consider infinitely many tasks parameterized by a continuous parameter, e.g., in *learning under non-stationarity* [3] where learning problems change over continuous time, *cost-sensitive learning* [4] where loss functions are asymmetric with continuous cost balance, and *quantile regression* [5] where the quantile is a continuous variable between zero and one. In order to handle these infinitely many parametrized tasks, we propose in this paper an extended formulation of MTL called *parametric-task learning (PTL)*.

The key contribution of this paper is to show that, for a certain class of PTL problems, the optimal common representation shared across infinitely many parameterized tasks can be obtainable. Specifically, we develop an alternating minimization algorithm à la [1, 2] for finding the entire continuum of solutions and the common feature subspace (or the common set of variables) among infinitely many parameterized tasks. Our algorithm exploits the fact that, for those classes of PTL problems, the path of task-wise solutions is piecewise-linear in the task parameter. We use the parametric programming technique [6, 7, 8, 9] for computing those piecewise linear solutions.

**Notations**: Let us denote by $\mathbb{R}$, $\mathbb{R}_+$, and $\mathbb{R}_{++}$ the set of real, nonnegative, and positive numbers, respectively, while we define $\mathbb{N}_n := \{1, \ldots, n\}$ for every natural number $n$. We denote by $\mathcal{S}_{++}^d$ the set of $d \times d$ positive definite matrices, and let $I(\cdot)$ be the indicator function.

## 2 Review of Multi-Task Learning (MTL)

In this section, we review an MTL method developed in [1, 2]. Let $\{(\boldsymbol{x}_i, y_i)\}_{i \in \mathbb{N}_n}$ be the set of $n$ training instances, where $\boldsymbol{x}_i \in \mathcal{X} \subseteq \mathbb{R}^d$ is the input and $y_i \in \mathcal{Y}$ is the output. We define $w_i(t) \in [0,1], t \in \mathbb{N}_T$ as the weight of the $i^{\text{th}}$ instance for the $t^{\text{th}}$ task, where $T$ is the number of tasks. We consider an affine model $f_t(\boldsymbol{x}) = \beta_{t,0} + \boldsymbol{\beta}_t^\top \boldsymbol{x}$ for each task, where $\beta_{t,0} \in \mathbb{R}$ and $\boldsymbol{\beta}_t \in \mathbb{R}^d$. For notational simplicity, we define augmented vectors $\tilde{\boldsymbol{\beta}} := (\beta_0, \beta_1, \ldots, \beta_d)^\top \in \mathbb{R}^{d+1}$ and $\tilde{\boldsymbol{x}} := (1, x_1, \ldots, x_d)^\top \in \mathbb{R}^{d+1}$, and write the affine model as $f_t(\boldsymbol{x}) = \tilde{\boldsymbol{\beta}}_t^\top \tilde{\boldsymbol{x}}$.

The multi-task feature learning method discussed in [1] is formulated as

$$\min_{\substack{\{\tilde{\boldsymbol{\beta}}_t\}_{t \in \mathbb{N}_T} \\ D \in \mathcal{S}_{++}^d, \text{tr}(D) \leq 1}} \sum_{t \in \mathbb{N}_T} \sum_{t \in \mathbb{N}_T} w_i(t) \ell_t(r(y_i, \tilde{\boldsymbol{\beta}}_t^\top \tilde{\boldsymbol{x}}_i)) + \frac{\gamma}{T} \sum_{t \in \mathbb{N}_T} \boldsymbol{\beta}_t^\top D^{-1} \boldsymbol{\beta}_t, \tag{1}$$

where $\text{tr}(D)$ is the trace of $D$, $\ell_t : \mathbb{R} \to \mathbb{R}_+$ is the loss function for the $t^{\text{th}}$ task incurred on the residual $r(y_i, \tilde{\boldsymbol{\beta}}_t^\top \tilde{\boldsymbol{x}}_i)$[1], and $\gamma > 0$ is the regularization parameter[2]. It was shown [1] that the problem (1) is equivalent to

$$\min_{\{\tilde{\boldsymbol{\beta}}_t\}_{t \in \mathbb{N}_T}} \sum_{t \in \mathbb{N}_T} \sum_{i \in \mathbb{N}_N} w_i(t) \ell_t(r(y_i, \tilde{\boldsymbol{\beta}}_t^\top \tilde{\boldsymbol{x}}_i)) + \frac{\gamma}{T} \|B\|_{\text{tr}}^2,$$

where $B$ is the $d \times T$ matrix whose $t^{\text{th}}$ column is given by the vector $\boldsymbol{\beta}_t$, and $\|B\|_{\text{tr}} := \text{tr}((BB^\top)^{1/2})$ is the *trace norm* of $B$. As shown in [10], the trace norm is the convex upper envelope of the rank of $B$, and (1) can be interpreted as the problem of finding a common feature subspace across $T$ tasks. This problem is often referred to as *multi-task feature learning*. If the matrix $D$ is restricted to be diagonal, the formulation (1) is reduced to *multi-task variable selection* [11, 12].

In order to solve the problem (1), the *alternating minimization algorithm* was suggested in [1] (see Algorithm 1). This algorithm alternately optimizes the task-wise solutions $\{\tilde{\boldsymbol{\beta}}_t\}_{t \in \mathbb{N}_T}$ and the common representation matrix $D$. It is worth noting that, when $D$ is fixed, each $\tilde{\boldsymbol{\beta}}_t$ can be independently optimized (Step 1). On the other hand, when $\{\tilde{\boldsymbol{\beta}}_t\}_{t \in \mathbb{N}_T}$ are fixed, the optimization of the matrix $D$ can be reduced to the minimization over $d$ eigenvalues $\lambda_1, \ldots, \lambda_d$ of the matrix $C := BB^\top$, and the optimal $D$ can be analytically computed (Step 2).

## 3 Parametric-Task Learning (PTL)

We consider the case where we have infinitely many tasks parametrized by a single continuous parameter. Let $\theta \in [\theta_{\text{L}}, \theta_{\text{U}}]$ be a continuous task parameter. Instead of the set of weights $w_i(t), t \in \mathbb{N}_T$, we consider a weight function $w_i : [\theta_{\text{L}}, \theta_{\text{U}}] \to [0, 1]$ for each instance $i \in \mathbb{N}_n$. In PTL, we learn a parameter vector $\tilde{\boldsymbol{\beta}}_\theta \in \mathbb{R}^{d+1}$ as a continuous function of the task parameter $\theta$:

$$\min_{\substack{\{\tilde{\boldsymbol{\beta}}_\theta\}_{\theta \in [\theta_{\text{L}}, \theta_{\text{U}}]} \\ D \in \mathcal{S}_{++}^d, \text{tr}(D) \leq 1}} \int_{\theta_{\text{L}}}^{\theta_{\text{U}}} \sum_{i \in \mathbb{N}_n} w_i(\theta) \, \ell_\theta(r(y_i, \tilde{\boldsymbol{\beta}}_\theta^\top \tilde{\boldsymbol{x}}_i)) \, d\theta + \gamma \int_{\theta_{\text{L}}}^{\theta_{\text{U}}} \boldsymbol{\beta}_\theta^\top D^{-1} \boldsymbol{\beta}_\theta \, d\theta, \tag{2}$$

where, note that, the loss function $\ell_\theta$ possibly depends on $\theta$.

As we will explain in the next section, the above PTL formulation is useful in various important machine learning scenarios including learning under non-stationarity, cost-sensitive learning, and

**Algorithm 1** ALTERNATING MINIMIZATION ALGORITHM FOR MTL [1]

1: **Input**: Data $\{(\boldsymbol{x}_i, y_i)\}_{i \in \mathbb{N}_n}$ and weights $\{w_i(t)\}_{i \in \mathbb{N}_n, t \in \mathbb{N}_T}$;
2: **Initialize**: $D \leftarrow I_d/d$ ($I_d$ is $d \times d$ identity matrix)
3: **while** convergence condition is not true **do**
4:     **Step 1**: **For** $t = 1, \ldots, T$ **do**

$$\tilde{\boldsymbol{\beta}}_t \leftarrow \arg\min_{\tilde{\boldsymbol{\beta}}} \sum_{i \in \mathbb{N}_n} w_i(t) \ell_t(r(y_i, \tilde{\boldsymbol{\beta}}^\top \tilde{\boldsymbol{x}}_i)) + \frac{\gamma}{T} \boldsymbol{\beta}^\top D^{-1} \boldsymbol{\beta}$$

5:     **Step 2**:

$$D \leftarrow \frac{C^{1/2}}{\text{tr}(C)^{1/2}} = \arg\min_{D \in \mathcal{S}^d_{++}, \text{tr}(D) \leq 1} \sum_{t \in \mathbb{N}_T} \boldsymbol{\beta}_t^\top D^{-1} \boldsymbol{\beta}_t,$$

    where $C := BB^\top$ whose $(j,k)^{\text{th}}$ element is defined as $C_{j,k} := \sum_{t \in \mathbb{N}_T} \beta_{tj} \beta_{tk}$.
6: **end while**
7: **Output**: $\{\tilde{\boldsymbol{\beta}}_t\}_{t \in \mathbb{N}_T}$ and $D$;

quantile regression. However, at first glance, the PTL optimization problem (2) seems computationally intractable since we need to find infinitely many task-wise solutions as well as the common feature subspace (or the common set of variables if $D$ is restricted to be diagonal) shared by infinitely many tasks.

Our key finding is that, for a certain class of PTL problems, when $D$ is fixed, the optimal path of the task-wise solutions $\tilde{\boldsymbol{\beta}}_\theta$ is shown to be piecewise-linear in $\theta$. By exploiting this piecewise-linearity, we can efficiently handle infinitely many parameterized tasks, and the optimal solutions of those class of PTL problems can be exactly computed.

In the following theorem, we prove that the task-wise solutions $\tilde{\boldsymbol{\beta}}_\theta$ is piecewise-linear in $\theta$ if the weight functions and the loss function satisfy certain conditions.

**Theorem 1** *For any $d \times d$ positive-definite matrix $D \in \mathcal{S}^d_{++}$, the optimal solution path of*

$$\tilde{\boldsymbol{\beta}}_\theta \leftarrow \arg\min_{\tilde{\boldsymbol{\beta}}} \sum_{i \in \mathbb{N}_n} w_i(\theta) \ell_\theta(r(y_i, \tilde{\boldsymbol{\beta}}^\top \tilde{\boldsymbol{x}}_i)) + \gamma \boldsymbol{\beta}^\top D^{-1} \boldsymbol{\beta} \tag{3}$$

*for $\theta \in [\theta_{\text{L}}, \theta_{\text{U}}]$ is written as a piecewise-linear function of $\theta$ if the residual $r(y, \tilde{\boldsymbol{\beta}}^\top \tilde{\boldsymbol{x}})$ can be written as an affine function of $\tilde{\boldsymbol{\beta}}$, and the weight functions $w_i : [\theta_{\text{L}}, \theta_{\text{U}}] \to [0, 1]$, $i \in \mathbb{N}_n$ and the loss function $\ell : \mathbb{R} \to \mathbb{R}_+$ satisfy either of the following conditions* **(a)** *or* **(b)***:*

**(a)** *All the weight functions are piecewise-linear functions, and the loss function is a convex piecewise-linear function which does not depend on $\theta$;*

**(b)** *All the weight functions are piecewise-constant functions, and the loss function is a convex piecewise-linear function which depends on $\theta$ in the following form:*

$$\ell_\theta(r) = \sum_{h \in \mathbb{N}_H} \max\{(a_h + b_h r)(c_h + d_h \theta), 0\}, \tag{4}$$

*where $H$ is a positive integer, and $a_h, b_h, c_h, d_h \in \mathbb{R}$ are constants such that $c_h + d_h \theta \geq 0$ for all $\theta \in [\theta_{\text{L}}, \theta_{\text{U}}]$.*

In the proof in Appendix A, we show that, if the weight functions and the loss function satisfy the conditions **(a)** or **(b)**, the problem (3) is reformulated as a *parametric quadratic program (parametric QP)*, where the parameter $\theta$ only appears in the linear term of the objective function. As shown, for example, in [9], the optimal solution path of this class of parametric QP has a piecewise-linear form.

If $\tilde{\boldsymbol{\beta}}_\theta$ is piecewise-linear in $\theta$, we can exactly compute the entire solution path by using parametric programming. In machine learning literature, parametric programming is often used in the context

---

**Algorithm 2** ALTERNATING MINIMIZATION ALGORITHM FOR PTL

---

1: **Input:** Data $\{(\boldsymbol{x}_i, y_i)\}_{i \in \mathbb{N}_n}$ and weight functions $w_i : [\theta_{\mathrm{L}}, \theta_{\mathrm{U}}] :\to [0, 1]$ for all $i \in \mathbb{N}_n$;
2: **Initialize**: $D \leftarrow I_d/d$ ($I_d$ is $d \times d$ identity matrix)
3: **while** convergence condition is not true **do**
4:     **Step 1**: **For** all the continuum of $\theta \in [\theta_{\mathrm{L}}, \theta_{\mathrm{U}}]$ **do**

$$\tilde{\boldsymbol{\beta}}_\theta \;\leftarrow\; \arg\min_{\tilde{\boldsymbol{\beta}}} \; \sum_{i \in \mathbb{N}_n} w_i(\theta) \ell_\theta(r(y_i, \tilde{\boldsymbol{\beta}}^\top \tilde{\boldsymbol{x}}_i)) + \gamma \boldsymbol{\beta}^\top D^{-1} \boldsymbol{\beta}$$

    by using parametric programming;
5:     **Step 2**:

$$D \;\leftarrow\; \frac{C^{1/2}}{\mathrm{tr}(C)^{1/2}} = \arg\min_{D \in \mathcal{S}_{++}^d, \mathrm{tr}(D) \le 1} \int_{\theta_{\mathrm{L}}}^{\theta_{\mathrm{U}}} \boldsymbol{\beta}_\theta^\top D^{-1} \boldsymbol{\beta}_\theta d\theta, \tag{5}$$

    where $(j, k)^{\mathrm{th}}$ element of $C \in \mathbb{R}^{d \times d}$ is defined as $C_{j,k} := \int_{\theta_{\mathrm{L}}}^{\theta_{\mathrm{U}}} \beta_{\theta,j} \beta_{\theta,k} d\theta$;
6: **end while**
7: **Output:** $\{\tilde{\boldsymbol{\beta}}_\theta\}$ for $\theta \in [\theta_{\mathrm{L}}, \theta_{\mathrm{U}}]$ and $D$;

---

of *regularization path-following* [13, 14, 15][3]. We start from the solution at $\theta = \theta_{\mathrm{L}}$, and follow the path of the optimal solutions while $\theta$ is continuously increased. This is efficiently conducted by exploiting the piecewise-linearity.

Our proposed algorithm for solving the PTL problem (2) is described in Algorithm 2, which is essentially a continuous version of the MTL algorithm shown in Algorithm 1. Note that, by exploiting the piecewise linearity of $\boldsymbol{\beta}_\theta$, we can compute the integral at Step 2 (Eq. (5)) in Algorithm 2.

Algorithm 2 can be changed to parametric-task variable selection if Step 2 is replaced with

$$D \;\leftarrow\; \mathrm{diag}(\lambda_1, \ldots, \lambda_d) \text{ where } \lambda_j = \frac{\sqrt{\int_{\theta_{\mathrm{L}}}^{\theta_{\mathrm{U}}} \beta_{\theta,j}^2 d\theta}}{\sum_{j' \in \mathbb{N}_d} \sqrt{\int_{\theta_{\mathrm{L}}}^{\theta_{\mathrm{U}}} \beta_{\theta,j'}^2 d\theta}} \text{ for all } j \in \mathbb{N}_d,$$

which can also be computed efficiently by exploiting the piecewise linearity of $\boldsymbol{\beta}_\theta$.

## 4 Examples of PTL Problems

In this section, we present three examples where our PTL formulation (2) is useful.

**Binary Classification Under Non-Stationarity**    Suppose that we observe $n$ training instances sequentially, and denote them as $\{(\boldsymbol{x}_i, y_i, \tau_i)\}_{i \in \mathbb{N}_n}$, where $\boldsymbol{x}_i \in \mathbb{R}^d$, $y_i \in \{-1, 1\}$, and $\tau_i$ is the time when the $i^{\mathrm{th}}$ instance is observed. Without loss of generality, we assume that $\tau_1 < \ldots < \tau_n$. Under non-stationarity, if we are requested to learn a classifier to predict the output for a test input $\boldsymbol{x}$ observed at time $\tau$, the training instances observed around time $\tau$ should have more influence on the classifier than others.

Let $w_i(\tau)$ denote the weight of the $i^{\mathrm{th}}$ instance when training a classifier for a test point at time $\tau$. We can for example use the following triangular weight function (see Figure1):

$$w_i(\tau) = \begin{cases} 1 + s^{-1}(\tau_i - \tau) & \text{if } \tau - s \le \tau_i < \tau, \\ 1 - s^{-1}(\tau_i - \tau) & \text{if } \tau \le \tau_i < \tau + s, \\ 0 & \text{otherwise,} \end{cases} \tag{6}$$

where $s > 0$ determines the width of the triangular time windows. The problem of training a classifier for time $\tau$ is then formulated as

$$\min_{\tilde{\boldsymbol{\beta}}} \sum_{i \in \mathbb{N}_n} w_i(\tau) \max(0, 1 - y_i \tilde{\boldsymbol{\beta}}^\top \tilde{\boldsymbol{x}}_i) + \gamma \|\boldsymbol{\beta}\|_2^2,$$

where we used the hinge loss.

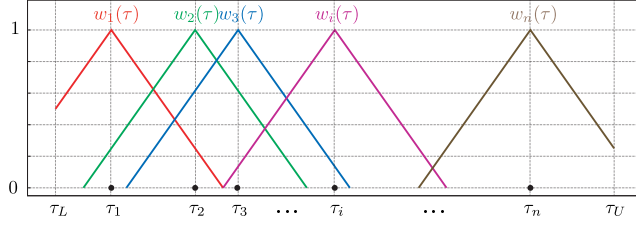

Figure 1: Examples of weight functions $\{w_i(\tau)\}_{i\in\mathbb{N}_n}$ in non-stationary time-series learning. Given a training instances $(\boldsymbol{x}_i, y_i)$ at time $\tau_i$ for $i = 1, \ldots, n$ under non-stationary condition, it is reasonable to use the weights $\{w_i(\tau)\}_{i\in\mathbb{N}_n}$ as shown here when we learn a classifier to predict the output of a test input at time $\tau$.

If we have the belief that a set of classifiers for different time should have some common structure, we can apply our PTL approach to this problem. If we consider a time interval $\tau \in [\tau_{\mathrm{L}}, \tau_{\mathrm{U}}]$, the parametric-task feature learning problem is formulated as

$$\min_{\substack{\{\tilde{\boldsymbol{\beta}}(\tau)\}_{\tau\in[\tau_{\mathrm{L}},\tau_{\mathrm{U}}]} \\ D\in\mathcal{S}_{++}^d,\mathrm{tr}(D)\leq 1}} \int_{\tau_{\mathrm{L}}}^{\tau_{\mathrm{U}}} \sum_{i\in\mathbb{N}_n} w_i(\tau)\,\max(0, 1 - y_i\tilde{\boldsymbol{\beta}}_\tau^\top\tilde{\boldsymbol{x}}_i)\,d\tau + \gamma \int_{\tau_{\mathrm{L}}}^{\tau_{\mathrm{U}}} \boldsymbol{\beta}_\tau^\top D^{-1}\boldsymbol{\beta}_\tau\,d\tau. \qquad (7)$$

Note that the problem (7) satisfies the condition **(a)** in Theorem 1.

**Joint Cost-Sensitive Learning**   Next, let us consider cost-sensitive binary classification. When the costs of false positives and false negatives are unequal, or when the numbers of positive and negative training instances are highly imbalanced, it is effective to use the *cost-sensitive learning* approach [16]. Suppose that we are given a set of training instances $\{(\boldsymbol{x}_i, y_i)\}_{i\in\mathbb{N}_n}$ with $\boldsymbol{x}_i \in \mathbb{R}^d$ and $y_i \in \{-1, 1\}$. If we know that the ratio of the false positive and false negative costs is approximately $\theta : (1 - \theta)$, it is reasonable to solve the following cost-sensitive SVM [17]:

$$\min_{\tilde{\boldsymbol{\beta}}} \sum_{i\in\mathbb{N}_n} w_i(\theta)\max(0, 1 - y_i\tilde{\boldsymbol{\beta}}^\top\tilde{\boldsymbol{x}}_i) + \gamma||\boldsymbol{\beta}||_2^2,$$

where the weight $w_i(\theta)$ is defined as

$$w_i(\theta) = \begin{cases} \theta & \text{if } y_i = -1, \\ 1 - \theta & \text{if } y_i = +1. \end{cases}$$

When the exact false positive and false negative costs in the test scenario are unknown [4], it is often desirable to train several cost-sensitive SVMs with different values of $\theta$. If we have the belief that a set of classifiers for different cost ratios should have some common structure, we can apply our PTL approach to this problem. If we consider an interval $\theta \in [\theta_{\mathrm{L}}, \theta_{\mathrm{U}}]$, $0 < \theta_{\mathrm{L}} < \theta_{\mathrm{U}} < 1$, the parametric-task feature learning problem is formulated as

$$\min_{\substack{\{\tilde{\boldsymbol{\beta}}_\theta\}_{\theta\in[\theta_{\mathrm{L}},\theta_{\mathrm{U}}]} \\ D\in\mathcal{S}_{++}^d,\mathrm{tr}(D)\leq 1}} \int_{\theta_{\mathrm{L}}}^{\theta_{\mathrm{U}}} \sum_{i\in\mathbb{N}_n} w_i(\theta)\,\max(0, 1 - y_i\tilde{\boldsymbol{\beta}}_\theta^\top\tilde{\boldsymbol{x}}_i)\,d\theta + \gamma \int_{\theta_{\mathrm{L}}}^{\theta_{\mathrm{U}}} \boldsymbol{\beta}_\theta^\top D^{-1}\boldsymbol{\beta}_\theta\,d\theta. \qquad (8)$$

The problem (8) also satisfies the condition **(a)** in Theorem 1. Figure 2 shows an example of joint cost-sensitive learning applied to a toy 2D binary classification problem.

**Joint Quantile Regression**   Given a set of training instances $\{(\boldsymbol{x}_i, y_i)\}_{i\in\mathbb{N}_n}$ with $\boldsymbol{x}_i \in \mathbb{R}^d$ and $y_i \in \mathbb{R}$ drawn from a joint distribution $P(\boldsymbol{X}, Y)$, *quantile regression* [19] is used to estimate the conditional $\tau^{\mathrm{th}}$ quantile $F_{Y|X=\boldsymbol{x}}^{-1}(\tau)$ as a function of $\boldsymbol{x}$, where $\tau \in (0, 1)$ and $F_{Y|X=\boldsymbol{x}}$ is the cumulative distribution function of the conditional distribution $P(Y|X = \boldsymbol{x})$. Jointly estimating multiple conditional quantile functions is often useful for exploring the stochastic relationship between $\boldsymbol{X}$ and $Y$ (see Section 5 for an example of joint quantile regression problems). Linear quantile regression along with $L_2$ regularization [20] at order $\tau \in (0, 1)$ is formulated as

$$\min_{\tilde{\boldsymbol{\beta}}} \sum_{i\in\mathbb{N}_n} \rho_\tau(y_i - \tilde{\boldsymbol{\beta}}^\top\tilde{\boldsymbol{x}}_i) + \gamma||\boldsymbol{\beta}||_2^2, \quad \rho_\tau(r) := \begin{cases} (1 - \tau)|r| & \text{if } r \leq 0, \\ \tau|r| & \text{if } r > 0. \end{cases}$$

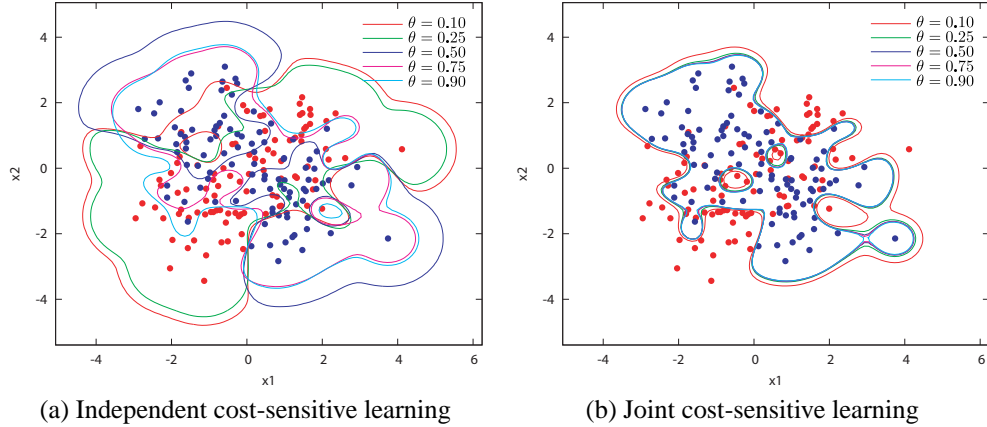

| (a) Independent cost-sensitive learning | (b) Joint cost-sensitive learning |

Figure 2: An example of joint cost-sensitive learning on 2D toy dataset (2D input $\boldsymbol{x}$ is expanded to $n$-dimension by radial basis functions centered on each $\boldsymbol{x}_i$). In each plot, the decision boundaries of five cost-sensitive SVMs ($\theta = 0.1, 0.25, 0.5, 0.75, 0.9$) are shown. (a) Left plot is the results obtained by independently training each cost-sensitive SVMs. (b) Right plot is the results obtained by jointly training infinitely many cost-sensitive SVMs for all the continuum of $\theta \in [0.05, 0.95]$ using the methodology we present in this paper (both are trained with the same regularization parameter $\gamma$). When independently trained, the inter-relationship among different cost-sensitive SVMs looks inconsistent (c.f., [18]).

If we have the belief that a family of quantile regressions at various $\tau \in (0, 1)$ have some common structure, we can apply our PTL framework to joint estimation of the family of quantile regressions This PTL problem satisfies the condition **(b)** in Theorem 1, and is written as

$$\min_{\substack{\{\boldsymbol{\beta}_\tau\}_{\tau \in (0,1)} \\ D \in \mathcal{S}_{++}^d, \mathrm{tr}(D) \leq 1}} \int_0^1 \sum_{i \in \mathbb{N}_n} \rho_\tau(y_i - \boldsymbol{\beta}_\tau^\top \boldsymbol{x}_i) d\tau + \gamma \int_0^1 \boldsymbol{\beta}_\tau^\top D^{-1} \boldsymbol{\beta}_\tau d\tau,$$

where we do not need any weighting and omit $w_i(\tau) = 1$ for all $i \in \mathbb{N}_n$ and $\tau \in [0, 1]$.

## 5 Numerical Illustrations

In this section, we illustrate various aspects of PTL with the three examples discussed in the previous section.

**Artificial Example for Learning under Non-stationarity** We first consider a simple artificial problem with non-stationarity, where the data generating mechanism gradually changes. We assume that our data generating mechanism produces the training set $\{(\boldsymbol{x}_i, y_i, \tau_i)\}_{i \in \mathbb{N}_n}$ with $n = 100$ as follows. For each $\tau_i \in \{0, 1\frac{2\pi}{n}, 2\frac{2\pi}{n}, \ldots, (n-1)\frac{2\pi}{n}\}$, the output $y_i$ is first determined as $y_i = 1$ if $i$ is odd, while $y_i = -1$ if $i$ is even. Then, $\boldsymbol{x}_i \in \mathbb{R}^d$ is generated as

$$x_{i1} \sim N(y_i \cos \tau_i, 1^2), \ x_{i2} \sim N(y_i \sin \tau_i, 1^2), \ x_{ij} \sim N(0, 1^2), \forall j \in \{3, \ldots, d\}, \qquad (9)$$

where $N(\mu, \sigma^2)$ is the normal distribution with mean $\mu$ and variance $\sigma^2$. Namely, only the first two dimensions of $\boldsymbol{x}$ differ in two classes, and the remaining $d - 2$ dimensions are considered as noise. In addition, according to the value of $\tau_i$, the means of the class-wise distributions in the first two dimensions gradually change. The data distributions of the first two dimensions for $\tau = 0, 0.5\pi, \pi, 1.5\pi$ are illustrated in Figure 3. Here, we applied our PT feature learning approach with triangular time windows in (6) with $s = 0.25\pi$. Figure 4 shows the mis-classification rate of PT feature learning (PTFL) and ordinary independent learning (IND) on a similarly generated test sample with size 1000. When the input dimension $d = 2$, there is no advantage for learning common features since these two input dimensions are important for classification. On the other hand, as $d$ increases, PT feature learning becomes more and more advantageous. Especially when the regularization parameter $\gamma$ is large, the independent learning approach is completely deteriorated as $d$ increases, while PTFL works reasonably well in all the setups.

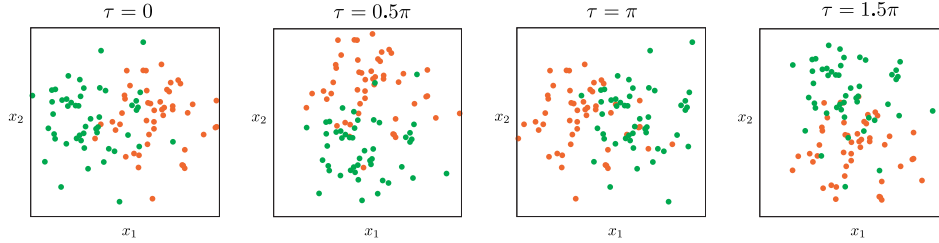

Figure 3: The first 2 input dimensions of artificial example at $\tau = 0, 0.5\pi, \pi, 1.5\pi$. The class-wise distributions in these two dimensions gradually change with $\tau \in [0, 2\pi]$.

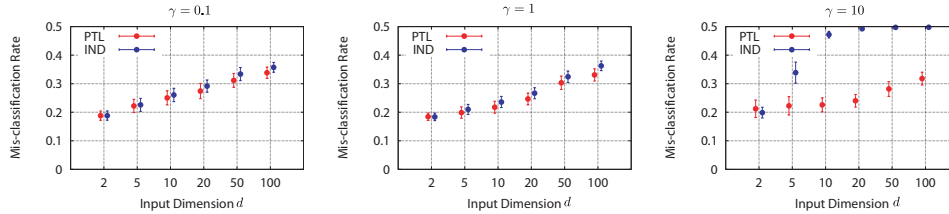

Figure 4: Experimental results on artificial example under non-stationarity. Mis-classification rate on test sample with size 1000 for various setups $d \in \{2, 5, 10, 20, 50, 100\}$ and $\gamma \in \{0.1, 1, 10\}$ are shown. The red symbols indicate the results of our PT feature learning (PTFL) whereas the blue symbols indicate ordinary independent learning (IND). The plotted are average (and standard deviation) over 100 replications with different random seeds. All the differences except $d = 2$ are statistically significant ($p < 0.01$).

**Joint Cost-Sensitive SVM Learning on Benchmark Datasets**    Here, we report the experimental results on joint cost-sensitive SVM learning discussed in Section 4. Although our main contribution is not just claiming favorable generalization properties of parametric task learning solutions, we compared, as an illustration, the generalization performances of PT feature learning (PTFL) and PT variable selection (PTVS) with the ordinary independent learning approach (IND). In PTFL and PTVS, we learned common feature subspaces and common sets of variables shared across the continuum of cost-sensitive SVM for $\theta \in [0.05, 0.95]$ for 10 benchmark datasets (see Table 1). In each data set, we divided the entire sample into training, validation, and test sets with almost equal size. The average test errors (and the standard deviation) of 10 different data splits are reported in Table 1. The total test errors for cost-sensitive SVMs with $\theta = 0.1, 0.2, \ldots, 0.9$ are defined as $\sum_{\theta \in \{0.1, \ldots, 0.9\}} \left( \theta \sum_{i:y_i=-1} I(f_\theta(\boldsymbol{x}_i) > 0) + (1 - \theta) \sum_{i:y_i=1} I(f_\theta(\boldsymbol{x}_i) \leq 0) \right)$, where $f_\theta$ is the trained SVM with the cost ratio $\theta$. Model selection was conducted by using the same criterion on validation sets. We see that, in most cases, PTFL or PTVS had better generalization performance than IND.

**Joint Quantile Regression**    Finally, we applied PT feature learning to joint quantile regression problems. Here, we took a slightly different approach from what was described in the previous section. Given a training set $\{(\boldsymbol{x}_i, y_i)\}_{i \in \mathbb{N}_n}$, we first estimated conditional mean function $E[Y | \boldsymbol{X} = \boldsymbol{x}]$ by least-square regression, and computed the residual $r_i := y_i - \hat{E}[Y | \boldsymbol{X} = \boldsymbol{x}_i]$, where $\hat{E}$ is the estimated conditional mean function. Then, we applied PT feature learning to $\{(\boldsymbol{x}_i, r_i)\}_{i \in \mathbb{N}_n}$, and estimated the conditional $\tau^{\text{th}}$ quantile function as $\hat{F}_{Y|\boldsymbol{X}=\boldsymbol{x}}^{-1}(\tau) := \hat{E}[Y | \boldsymbol{X} = \boldsymbol{x}_i] + \hat{f}_{\text{res}}(\boldsymbol{x}|\tau)$, where $\hat{f}_{\text{res}}(\cdot|\tau)$ is the estimated $\tau^{\text{th}}$ quantile regression fitted to the residuals.

When multiple quantile regressions with different $\tau$s are independently learned, we often encounter a notorious problem known as *quantile crossing* (see Section 2.5 in [5]). For example, in Figure 5(a), some of the estimated conditional quantile functions *cross* each other (which never happens in the true conditional quantile functions). One possible approach to mitigate this problem is to assume a model on the heteroscedastic structure. In the simplest case, if we assume that the data is *homoscedastic* (i.e., the conditional distribution $P(Y|\boldsymbol{x})$ does not depend on $\boldsymbol{x}$ except its location),

Table 1: Average (and standard deviation) of test errors obtained by joint cost-sensitive SVMs on benchmark datasets. $n$ is the sample size, $d$ is the input dimension, `Ind` indicates the results when each cost-sensitive SVM was trained independently, while `PTFL` and `PTVS` indicate the results from PT feature learning and PT feature selection, respectively. The bold numbers in the table indicate the best performance among three methods.

| Data Name | $n$ | $d$ | Ind | PTFL | PTVS |
|---|---|---|---|---|---|
| *Parkinson* | 195 | 20 | 32.30 (10.60) | **30.21 (9.09)** | 30.25 (8.53) |
| *Breast Cancer Diagnostic* | 569 | 30 | 20.36 (7.77) | **18.49 (6.15)** | 19.46 (5.89) |
| *Breast Cancer Prognostic* | 194 | 33 | 48.97 (12.92) | 49.28 ( 9.83) | **48.68 (5.89)** |
| *Australian* | 690 | 14 | 117.97 (22.97) | **106.25 (12.66)** | 111.22 (15.95) |
| *Diabetes* | 768 | 8 | 185.90 (21.13) | 179.89 (16.31) | **175.95 (16.26)** |
| *Fourclass* | 862 | 2 | 181.69 (22.13) | 179.30 (14.25) | **178.67 (19.24)** |
| *Germen* | 1000 | 24 | 242.21 (18.35) | **219.66 (16.22)** | 237.20 (15.78) |
| *Splice* | 1000 | 60 | 179.80 (24.22) | **151.69 (18.02)** | 183.54 (21.27) |
| *SVM Guide* | 300 | 10 | 175.70 (15.55) | **170.16 (9.99)** | 179.76 (14.76) |
| D*Vowel* | 528 | 10 | **175.16 (13.78)** | 175.74 (9.37) | 175.50 (7.38) |

quantile regressions at different $\tau$s can be obtained by just vertically shifting other quantile regression function (see Figure 5(f)).

Our PT feature learning approach, when applied to the joint quantile regression problem, allows us to *interpolate* these two extreme cases. Figure 5 shows a joint QR example on the bone mineral density (BMD) data [21]. We applied our approach after expanding univariate input $x$ to a $d = 5$ dimensional vector by using evenly allocated RBFs. When (a) $\gamma \to 0$, our approach is identical with independently estimating each quantile regression, while it coincides with homoscedastic case when (f) $\gamma \to \infty$. In our experience, the best solution is usually found somewhere between these two extremes: in this example, (d) $\gamma = 5$ was chosen as the best model by 10-fold cross-validation.

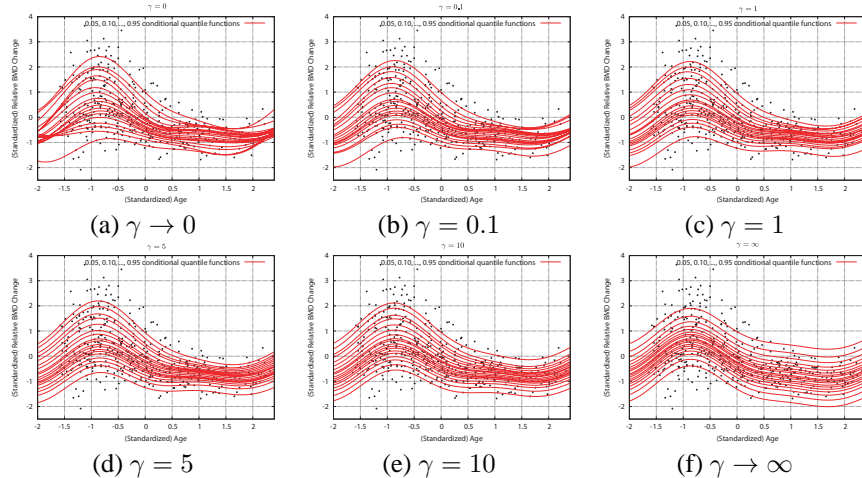

Figure 5: Joint quantile regression examples on BMD data [21] for six different $\gamma$s.

## 6  Conclusions

In this paper, we introduced parametric-task learning (PTL) approach that can systematically handle infinitely many tasks parameterized by a continuous parameter. We illustrated the usefulness of this approach by providing three examples that can be naturally formulated as PTL. We believe that there are many other practical problems that falls into this PTL framework.

**Acknowledgments**

The authors thank the reviewers for fruitful comments. IT, MS, and SN thank the support from MEXT Kakenhi 23700165, JST CREST Program, MEXT Kakenhi 23120004, respectively.

## Footnotes

[1]For example, $r(y_i, \tilde{\boldsymbol{\beta}}_t^\top \tilde{\boldsymbol{x}}_i) = (y_i - \tilde{\boldsymbol{\beta}}^\top \tilde{\boldsymbol{x}}_i)^2$ for regression problems with $y_i \in \mathbb{R}$, while $r(y_i, \tilde{\boldsymbol{\beta}}_t^\top \tilde{\boldsymbol{x}}_i) = 1 - y_i \tilde{\boldsymbol{\beta}}_t^\top \tilde{\boldsymbol{x}}_i$ for binary classification problems with $y_i \in \{-1, 1\}$.

[2]In [1], $w_i(t)$ takes either 1 or 0. It takes 1 only if the $i^{\text{th}}$ instance is used in the $t^{\text{th}}$ task. We slightly generalize the setup so that each instance can be used in multiple tasks with different weights.

[3]In regularization path-following, one computes the optimal solution path w.r.t. the regularization parameter, whereas we compute the optimal solution path w.r.t. the task parameter $\theta$.

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
