[Supplementary Material]

## Appendix A: Proof of Theorem 1

First, we prove the claim for the condition **(a)**. Let us divide the interval $[\theta_\mathrm{L}, \theta_\mathrm{U}]$ into finite number of segments so that, within each segment, the weight vector $\boldsymbol{w}(\theta) := (w_1(\theta), \ldots, w_n(\theta))^\top \in [0, 1]^n$ changes linearly with $\theta$, and denote the breakpoints of those segments as $\theta_\mathrm{L} = \theta_0 < \theta_1 < \ldots < \theta_s < \ldots < \theta_S = \theta_\mathrm{U}$, where $S$ is the number of those segments.

Then, consider a segment defined on $\theta \in [\theta_s, \theta_{s+1}]$, $s \in \{0, \ldots, S-1\}$, and denote the weight vectors at $\theta_s$ and $\theta_{s+1}$ as $\boldsymbol{w}(\theta_s)$ and $\boldsymbol{w}(\theta_{s+1})$, respectively. The problem of computing the solution path within this segment is written as the following parametric optimization problem

$$\tilde{\boldsymbol{\beta}}_\mu \leftarrow \arg\min_{\tilde{\boldsymbol{\beta}}} \sum_{i \in \mathbb{N}_n} ((1-\mu)w_i(\theta_s) + \mu w_i(\theta_{s+1}))\ell_{(1-\mu)\theta_s + \mu\theta_{s+1}}(r(y_i, \tilde{\boldsymbol{\beta}}^\top \tilde{\boldsymbol{x}}_i)) + \gamma \boldsymbol{\beta}^\top D^{-1} \boldsymbol{\beta} \tag{10}$$

for $\mu \in [0, 1]$.

Since the loss function $\ell_\theta$ does not depend on $\theta$ and is convex piecewise-linear in $r$, we can write $\ell_\theta$ as

$$\ell_\theta(r(y_i, \tilde{\boldsymbol{\beta}}^\top \tilde{\boldsymbol{x}}_i)) = \sum_{h \in \mathbb{N}_H} \max\{\phi_{ih} + \psi_{ih} \cdot r(y_i, \tilde{\boldsymbol{\beta}}^\top \tilde{\boldsymbol{x}}_i)\},$$

where $\phi_{ih}, \psi_{ih} \in \mathbb{R}, (i, h) \in \mathbb{N}_n \times \mathbb{N}_H$ are constants, and $H$ is the number of pieces of the piecewise-linear loss function $\ell_\theta$ (see, for example, section 4.3.1 of [22]).

Using slack variables $\boldsymbol{\xi} = (\xi_1, \ldots, \xi_n) \in \mathbb{R}^n$, the parametric programming problem in (10) is rewritten as

$$\{\tilde{\boldsymbol{\beta}}_\mu, \boldsymbol{\xi}_\mu\} \leftarrow \arg\min_{\tilde{\boldsymbol{\beta}}, \boldsymbol{\xi}} \quad ((1-\mu)\boldsymbol{w}(\theta_s) + \mu\boldsymbol{w}(\theta_{s+1}))^\top \boldsymbol{\xi} + \gamma \boldsymbol{\beta}^\top D^{-1} \boldsymbol{\beta}$$

$$\text{s.t.} \qquad \xi_i \geq \phi_{ih} + \psi_{ih} \cdot r(y_i, \tilde{\boldsymbol{\beta}}^\top \tilde{\boldsymbol{x}}_i) \text{ for all } (i, h) \in \mathbb{N}_n \times \mathbb{N}_H \tag{11}$$

with respect to $\mu \in [0, 1]$. The problem (11) belongs to the class of *parametric QP* (note that, when $\mu$ is fixed, the problem (11) is quadratic program with respect to $\tilde{\boldsymbol{\beta}}$ and $\boldsymbol{\xi}$, which has a quadratic objective function and a set of linear constraints.). As shown, for example, in [6, 9], a parametric quadratic program which contains the parameter ($\mu$) in the linear term of the quadratic objective function are shown to have a solution path in piecewise-linear form.

Similarly for the condition **(b)**, we consider a segment defined on $\theta \in [\theta_t, \theta_{s+1}]$, $s \in \{0, \ldots, S-1\}$, in which the weight vector $\boldsymbol{w}(\theta)$ is constant (and thus omitted hereafter). Using slack variables $\xi_{ih}$ for $i \in \mathbb{N}_n$ and $h \in \mathbb{N}_H$

$$\min_{\tilde{\boldsymbol{\beta}}} \sum_{i \in \mathbb{N}_n} \sum_{h \in \mathbb{N}_H} \max\{(a_h + b_h \cdot r(y_i, \tilde{\boldsymbol{\beta}}^\top \tilde{\boldsymbol{x}}_i))(c_h + d_h\theta), 0\} + \gamma \boldsymbol{\beta}^\top D^{-1} \boldsymbol{\beta}$$

$$\Leftrightarrow \min_{\tilde{\boldsymbol{\beta}}} \sum_{h \in \mathbb{N}_H} (c_h + d_h\theta) \sum_{i \in \mathbb{N}_n} \max\{(a_h + b_h \cdot r(y_i, \tilde{\boldsymbol{\beta}}^\top \tilde{\boldsymbol{x}}_i)), 0\} + \gamma \boldsymbol{\beta}^\top D^{-1} \boldsymbol{\beta}$$

$$\Leftrightarrow \min_{\tilde{\boldsymbol{\beta}}, \boldsymbol{\xi}} \sum_{h \in \mathbb{N}_H} (c_h + d_h\theta) \sum_{i \in \mathbb{N}_n} \xi_{ih} + \gamma \boldsymbol{\beta}^\top D^{-1} \boldsymbol{\beta}$$

$$\text{s.t. } \xi_{ih} \geq a_h + b_h \cdot r(y_i, \tilde{\boldsymbol{\beta}}^\top \tilde{\boldsymbol{x}}_i), \ \xi_{ih} \geq 0 \ \forall \ (i, h) \in \mathbb{N}_n \times \mathbb{N}_H.$$

The parametric programming problem in Theorem 1 **(b)** is thus written as

$$\{\tilde{\boldsymbol{\beta}}_\theta, \boldsymbol{\xi}_\theta\} \leftarrow \min_{\tilde{\boldsymbol{\beta}}, \boldsymbol{\xi}} \sum_{h \in \mathbb{N}_H} (c_h + d_h\theta) \sum_{i \in \mathbb{N}_n} \xi_{ih} + \gamma \boldsymbol{\beta}^\top D^{-1} \boldsymbol{\beta}$$

$$\text{s.t. } \xi_{ih} \geq a_h + b_h \cdot r(y_i, \tilde{\boldsymbol{\beta}}^\top \tilde{\boldsymbol{x}}_i), \ \xi_{ih} \geq 0 \ \forall \ (i, h) \in \mathbb{N}_n \times \mathbb{N}_H$$

for $\theta \in [\theta_s, \theta_{s+1}]$, and it also belongs to parametric QP, meaning that the optimal solution path is shown to be piecewise linear in $\theta$. $\qquad\square$