[Reviews · NeurIPS 2013]

Submitted by Assigned_Reviewer_4

This paper extends the popular "multi-task feature learning" formulation to the infinite task scenario:
1) Instead of having a finite number task, the problem has an infinite number of tasks characterized by a continuous task parameter \theta (e.g., \theta could be time t).
2) Instead of having each example belong to only one of the finite tasks, each example can belong to the entire class of infinite tasks where its weight in a task is defined by w(\theta), where \theta is the task parameter.
3) The author shows that it's possible to solve this infinite-task feature learning problem because, under certain conditions, the model parameters \beta(\theta) is piece-wise linear on the task parameter \theta.


The Originality, Quality, Clarity and Significance are reviewed as follows.

Originality:
- The idea of having infinite tasks characterized by a task parameter is novel and interesting.
- The authors also discussed and studied a few real-world scenarios, where the infinite-task formulation looks a natural solution.

Clarity:
- Overall the paper is well written. Examples used to illustrate the applications of infinite-task formulation is clear and interesting.
- Discussion/review on related work can be improved. One way to view the presented work is to impose a strong known task structure (characterized by a continuous task parameter) on multi-task feature learning formula (in order to handle infinite tasks). In this sense, the author should discuss previous work that tries to infer (unknown) task structure and incorporate it into multi-task feature learning, e.g., "Learning Multiple Tasks with a Sparse Matrix-Normal Penalty" at NIPS 2010. Can we combine the strength of inferring unknown task structure and handling infinite tasks?

Quality:
- The paper only extend a very specific type of multi-learning method -- the multi-task feature learning -- to the infinite-task case.
- The empirical study part is relatively weak, mostly based on simulation or small toy data. I like to see the application of the proposed formulation to some significant problem with real-world impact.

Significance:
- The infinite-task learning idea is interesting and has potential applications.
Summary: This paper extends "multi-task feature learning" to infinite-task cases, where an infinite number of tasks are characterized by a continuous task parameter and each example is (more or less) belong to all tasks. The idea is interesting and has many potential applications. Discussion on related work can be improved, and the authors may also discuss extending more multi-task learning methods to infinite-task cases.

Submitted by Assigned_Reviewer_5

This paper presents a parametric task learning whose formulation is motivated by multi-task feature learning. I don’t think the proposed method is a generalization of multi-task learning since most instances of parametric task learning is not multi-task learning problems.

Due to computational reason, the type of loss functions used in PTL is very limited and this prevents its application to many problems.

The authors should present the detailed optimization procedure to the step 1 in Algorithm 2 since many readers may not be familiar with parametric quadratic programming.

The experiments are not very satisfactory. Experiments on toy data for non-stationary are not enough. The authors are encouraged to do more experiments on real-world datasets. In experiments on cost-sensitive learning, the authors don’t compare their method with the benchmark cost-sensitive methods and so I cannot say the proposed method is good in terms of the generalization performance. For the joint quantile regression, the approach as described from line 365 to 372 is different from the method described in section 4. Why use this alternative approach? Moreover, the authors don’t compare with other quantile regression methods.
Summary: This paper presents a parametric task learning motivated by multi-task feature learning. The experiments are not very satisfactory.

Submitted by Assigned_Reviewer_6

The authors consider problems where tasks are related by a continuous parameter. They show that jointly optimizing over the range of the parameter can result in more consistent classifiers/regressors and thus better performance.

The paper is clearly written, and the main contribution is showing that under certain assumptions, the entire path of solutions over the range of the parameter can be computed via a parametric QP.

The consistency/quality of the solutions is demonstrated on a non-stationary regression problem, cost-senstive SVM problem, and quantile regression. (Note: for the SVM benchmarks, it seems like the standard deviations overlap in Table 1. It's probably honest to point that out.)
Summary: Useful contribution, clean exposition.
Author Feedback

Author rebuttal: Dear Reviewers,

Thank you for fruitful comments. Please find our answers to the questions below.

Reply to Reviewer 4

[4-1] Discussion/review on related work can be improved.

We agree with the reviewer in that an alternative approach to learning infinitely many related tasks is to impose or learn a restricted (e.g., parametric) model of task structure. In this view, the novelty of our work is that common shared structure for infinitely many tasks can be identified in a fully nonparametric way with the help of parametric programming. We will discuss this viewpoint and argue the high-level difference with the suggested (and possibly other) related works in the final version.

[4-2] The paper only extends a very specific type of MTL method

Our high-level idea for handling infinitely many tasks is the use of parametric programming. Although we restricted our attention to the problems that can be cast into "piecewise-linear" parametric programming where the computation is quite efficient, the same idea can be naturally extended to more general nonlinear parametric programming situations. In particular, recently proposed approximate parametric programming (with approximation guarantee) is useful for such generalization. In the final version, we will describe our high-level idea, and discuss the possible generalization to other types of MTL methods.

[4-3] The empirical study part is relatively weak

Our main goal is to introduce the parametric task learning (PTL) framework and show that there are many problems which can be naturally formulated as PTL. That is why we provided three simple examples rather than focusing on a single particular application in detail. We will discuss possible real-world applications of our PTL framework in the final version, including the use of joint quantile regression in financial data analysis and other fields.

Reply to Reviewer 5

[5-1] Most instances of parametric task learning is not multi-task learning problems.

We use the term multi-task learning (MTL) in a broad sense to refer to the situations where there are many tasks which are expected to share some common representation. PTL is a generalization of MTL to infinitely many tasks. There are many practical problems that can be naturally cast into the PTL framework (e.g., see [4-3] above).

[5-2] The type of loss functions used in PTL is limited

See our answer to comment [4-2] from Rev 4.

[5-3] The authors should present the detailed optimization procedure

We will describe the detail procedure in step 1 in the appendix of the final version. Thanks for the suggestion.

[5-4] The experiments are not very satisfactory.

See our answer to comment [4-3] from Rev 4.

[5-5] comparison with benchmark cost-sensitive methods

Sorry for the confusion. In the experiment, we compared our PTL approach with cost-sensitive SVM ('Ind' indicates cost-sensitive SVM). We will clarify this point in the final version.

[5-6] QR approach in line 365-372 is different from the method described in section 4

Sorry for confusion. In practical applications of quantile regression, it is usually recommended to separately model the homoscedastic part (i.e., conditional mean function E[Y|x]) and the heteroscedastic part, because each part should be often penalized with different magnitudes. In our manuscript, we simplified the description and avoided application-specific details in Section 4 since our objective was to provide examples of PTL. We will clarify this issue in Section 6 of the final version.

Reply to Reviewer 6

[6-1] Standard deviations overlap in Table 1

Thank you for pointing out this. The differences are sometimes not statistically significant. We will clarify this fact in the final version.